# HIV-HCV Incidence in Low-Wage Agricultural Migrant Workers Living in Ghettos in Apulia Region, Italy: A Multicenter Cross Sectional Study

**DOI:** 10.3390/v15010249

**Published:** 2023-01-15

**Authors:** Valentina Totaro, Giulia Patti, Francesco Vladimiro Segala, Renato Laforgia, Lucia Raho, Carmine Falanga, Marcella Schiavone, Luísa Frallonardo, Gianfranco Giorgio Panico, Vito Spada, Laura De Santis, Carmen Pellegrino, Roberta Papagni, Angelo D’Argenio, Roberta Novara, Claudia Marotta, Nicole Laforgia, Davide Fiore Bavaro, Giovanni Putoto, Annalisa Saracino, Francesco Di Gennaro

**Affiliations:** 1Clinic of Infectious Diseases, Department of Precision and Regenerative Medicine and Ionian Area (DiMePRe-J), University of Bari Aldo Moro, 70124 Bari, Italy; 2Doctors with Africa CUAMM, 70123 Bari, Italy; 3ANLAIDS Sezione Lombardia, 20124 Milan, Italy; 4Operational Research Unit, Doctors with Africa CUAMM, 35121 Padua, Italy

**Keywords:** migrant, health status, HIV, HCV, ghettos, Italy

## Abstract

Migrant populations are more susceptible to viral hepatitis and HIV due to the epidemiology from their country of origin or their social vulnerability when they arrive in Europe. The aims of the study are to explore the incidence of HIV and HCV in low-wage agricultural migrant workers and their knowledge, attitude, and practice with regard to HIV and HCV, as well as their sexual behaviour and risk factors. As part of the mobile clinic services, we performed a screening campaign for HIV-HCV involving migrants living in three Apulian establishments. Results: Between January 2020 and April 2021, 309 migrants (n. 272, 88% male, mean age 28.5 years) were enrolled in the study. Most of the migrants interviewed (n = 297, 96%) reported a stopover in Libya during their trip to Italy. Only 0.9% (n. 3) of migrants reported having been tested for HCV, while 30.7% (n. 95) reported being tested for HIV. Furthermore, screening tests found four migrants (1.3%) to be HIV positive and nine (2.9%) to be HCV positive. The median knowledge score was 1 (IQR 0-3; maximum score: 6 points) for HCV and 3 (IQR 1-4; maximum score: 7 points) for HIV and low use of condoms was 5% (n. 16), while more than 95% show an attitude score of 5 (IQR 5-6; maximum score:6 points) on HIV-HCV education campaigns. In a multivariate analysis, being male (OR = 1.72; 95% CI 1.28–1.92), being single (OR = 1.63; 95% CI 1.20–2.03), being of low educational status (OR = 2.09; 95% CI 1.29–2.21), living in shantytowns for >12 months (OR = 1.95; 95% CI 1.25–2.55), and originating from the African continent (OR = 1.43; 95% CI 1.28–2.01) are significant predictors of poor knowledge on HCV. Our data show low knowledge, especially of HCV, confirming migrants as a population with a higher risk of infection. To develop education programmes, integrated care and screening among migrants could be an effective strategy, considering the high attitude toward these items shown in our study.

## 1. Introduction

Despite progress in controlling the spread of infectious diseases, HIV and HCV still represent a major global health problem. In fact, in 2021, there were an estimated 38.4 million HIV-positive individuals worldwide, two thirds of whom lived in the WHO (World Health Organisation) African Region; 1.5 million people contracted HIV, and 650,000 people died from HIV-related causes [1]. Approximately 1.5 million new cases of the hepatitis C virus are reported each year, with an estimated 58 million people worldwide, 3.2 million of whom are children and adolescents, carrying a chronic infection. According to the WHO, 290,000 people died from hepatitis C in 2019, primarily from cirrhosis and hepatocellular carcinoma (primary liver cancer). The highest burden of HCV disease is in the Eastern Mediterranean and European regions, the South-East Asia region, the Western Pacific region, and the African region [2]. Europe is a major migrant destination, with most migrants from HIV and HCV endemic countries entering the continent through Italy [3]. Based on UNHCR arrival registration data, about 400,000 migrants have arrived in Italy in recent years. There have already been about 70,000 sea arrivals since January 2022 [4]. Once in Italy, these individuals sadly often have a low income, most frequently due to casual day-to-day work, broken family ties, and no fixed address [5]. In Italy, there are 500,000 migrants employed in agriculture. Many of them reside in shantytowns, incorrectly referred to as “ghettos”, which are remote from urban centres with poor hygienic conditions and lack access to water, electricity, or health services. The majority of them are from low and middle income countries such as those of sub-Saharan Africa, Eastern Europe and Asia. Their surroundings are miserable. Agricultural workers are a class of exploited employees since there are no laws or protections for them as workers, and their pay is inadequate for the kind of job they do and the hours they work. It is estimated that there are 50–70 shantytowns that host 100,000 migrant laborers [6]. The health of refugees and migrants is influenced by the conditions in their countries of origin, while traveling, and in their host communities. Despite the existence of a universal health care system and legislation providing for the health of migrants, they have limited to no access to basic medical care. These and other sub-optimal health determinants, such as education, income, and housing, compounded by linguistic, cultural, and legal barriers, are causes of poor health outcomes and the spread of infectious disease [7,8]. The majority of migrants are unaware of their HBV, HCV, and HIV status, although they come from endemic countries [5]. Consequently, certain migrant subgroups are more susceptible to viral hepatitis and HIV due to their prior exposure to risk factors owing to the conditions of social marginality in which they live in Italy and Europe; they are also more likely to go undiagnosed than the general population of their host country and they are more likely not to be supported by the healthcare system due to complex social factors, language barriers, social vulnerability or other barriers to their inclusion in the health system [9,10,11]. These subgroups should thus be informed, screened, and provided with specialised treatment. The primary objectives of the study were to define the incidence of HIV and HCV among migrant agricultural workers, as well as to determine their knowledge, attitudes, behaviors, and risk factors for HIV and HCV.

## 2. Materials and Methods

### 2.1. Study Setting, Design and Population

According to a regional programme and local and regional institutions, a cross-sectional, multicenter HIV-HCV screening was conducted from 10 January 2020, to 20 April 2021 in three ghettos in Apulia with HIV/HCV saliva testing. We formed a multidisciplinary team consisting of at least one specialist physician in infectious diseases, a medical resident in infectious diseases from the University of Bari, nurses, a cultural mediator, and a number of volunteers from Doctors with Africa CUAMM, which has been working in the ghettos since 2015 and also provides logistical support.

We included in these three Apulian establishments in our study:The Ghetto Pista in Borgo Mezzanone, in the province of Manfredonia, with an estimated 2500 migrant workers, from Africa and Asian;“Casa Sankara” and “Arena”, establishments organized by the Apulia region for the agricultural worker population, which is predominantly from African nations;The Gran Ghetto, located in Rignano Garganico, where an estimated 1000 migrants from African countries lived.

The eligible population comprised all individuals present in these establishments during the study period. In this research, no exclusion criteria were utilized.

#### Questionnaires

The development of the questionnaire was informed by a literature review and administered through a face-to-face interview conducted by a medical resident in infectious diseases and a nurse, with the support of a cultural and linguistic mediator. It was made of questions divided into four sections: (I) socio-demographic information (age, marital status, education, occupation typology of work contract, documentation to remain in Italy); (II) sexual habits (condom use, smoking habits, etc.); (III) information on HIV-HCV status (previous test if performed); and (IV) survey KAP (knowledge, attitude, and practice), with a 5-point Likert-style scale on HIV and HCV. To guarantee the confidentiality of the data, before to conducting the interview, informed consent was obtained, and the objectives and methods of the study were explained. After obtaining informed consent, an OraQuick rapid antibody test for HIV-1 and HIV-2 and an OraQuick anti-HCV test were performed. The collected data were entered into a dedicated online platform (Kobotool), and a quality control check of the data entry was performed before data analysis.

### 2.2. Statistical Analysis

A descriptive analysis was performed to define the distribution of the characteristics of the sample, and an χ2 test (with Fisher’s correction if fewer than five cases were present in a cell) was applied for categorical variables. An analysis of determinants of knowledge on HCV was conducted through the construction of multiple logistic regression models. The variable “knowledge on HCV” was collapsed into two levels: a high level of knowledge on HCV was attributed to respondents who provided correct responses to at least three of the six questions included in the knowledge section of the questionnaire, while a low level of knowledge on HCV was defined as those who provided only two correct answers out of six.

Covariates included in the models were: type of educational status (<8 ys vs. >8ys); participants’ sex; participants’ age; marital status; continent of origin (Asia vs. Africa), whether they had a family doctor, type of work (regular vs. irregular), previous HIV test, previous HCV test, comorbidity, and main area of work (surgical, clinical, non-clinical). Multiple logistic regression models were built. Each variable was examined by univariate analysis using the appropriate statistical test (Student’s *t*-test or χ2 test) and was included in the model when the *p*-value was <0.25. Subsequently, multivariate logistic regression with backward elimination of any variable that did not contribute to the model on the grounds of the likelihood ratio test (cut-off, *p* = 0.05) was performed. Adjusted odds ratios and 95% confidence intervals were calculated. All statistical calculations were performed using Stata v.15.0 (Stata Corp., College Station, TX, USA)

## 3. Results

Between 10 January 2020, and 20 April 2021, 309 migrants (n = 272, 88% male, mean age 28.5 years, IQR 18-61), residents of the ghettos of the Apulian region (n = 69 in Gran Ghetto, n = 175 in Ghetto Pista, and n = 65 in Casa Sankara), were enrolled in the study. Most of the migrants interviewed (n = 297, 96%) reported a stopover in Libya during their trip to Italy, 77% came from African countries, with an average length of stay in Italy of about 55 months, and a stay in the ghetto of more than 18 months. Only 38% (n =117) had a regular work permit; 29% (n = 89) were family doctors, with roughly 70% (n = 211) having completed more than 8 years of school. Furthermore, previous HIV and HCV tests had been carried out in 30.7% of cases (no. 95) for HIV and in only 0.9% of cases (no. 3) had testing been undertaken for HCV. Other sociodemographic information collected is reported in Table 1.

Knowledge of HIV and HCV is shown in Table 2. The median knowledge score was 1 (IQR 0-3; maximum score: 6 points) for HCV and 3 (IQR 1-4; maximum score: 7 points) for HIV. Five percent (N.15) of the population under examination strongly agree and 10% (N.31) agree that HIV can be transmitted through kissing a person living with HIV. Approximately 50% (195/309) of the population strongly agree that HCV can be transmitted through kissing a person living with HCV, while only 20% (n. 61) agree that HCV is transmitted through unprotected sex, while about forty-one per cent (n. 127) believe HCV causes liver cancer. With regard to HIV, 69% (n. 212) of the population under examination strongly agree/or agree that HIV is transmitted by contamination with infected blood, tattooing, and syringe use, and only 16% (n. 50) believe that, if untreated, HIV could be transmitted from mother to child during birth or pregnancy. The full series of questions and responses are shown in Table 2.

With regard to the attitudes towards HIV and HCV, 55% (n. 170) of the population under examination agree and 37% (n. 114) strongly agree with receiving HIV treatment in case of infection; 60% (n. 185) of the population under examination agree and 34% (n. 105) strongly agree with taking HCV treatment. Our study’s total population (100%, n. 309) supports education and screening campaigns (considering that 85% (n. 263) agree and 15% (n. 46) strongly agree with HIV education campaigns; and 80% (n. 247) agree and 20% (n. 62) strongly agree with HCV education campaigns; 77% (n. 238) agree and 23% (n. 71) strongly agree for HIV screening campaigns. Only a very small percentage would avoid relationships with infected people. Table 3 shows all of the questions and responses concerning attitudes towards HIV/.

In practice, only 5% of men (n. 15) always have sex with a condom, and 50% (n. 155) do not recommend the use of condoms during sex. On the other hand, almost 100% of never-used syringes were already used, as showed in Table 4. 

In a multivariate analysis, shown in Table 5, being male (OR = 1.72; 95% CI 1.28–1.92), being single (OR = 1.63; 95% CI 1.20–2.03), being of low educational status (OR = 2.09; 95% CI 1.29–2.21), living in the ghettos for >12 months (OR = 1.95; 95% CI 1.25–2.55), and having an African origin (OR = 1.43; 95% CI 1.28–2.01) are significant predictors of poor knowledge of HCV. On the contrary, having regular work (OR = 0.64; 95% CI 0.39–0.83), having a family doctor (OR = 0.24, 95%CI 0.18–0.73), and having performed a previous HIV/HCV test (OR = 0.59; 95% CI 0.10–0.90) are indicative of a low knowledge of HCV.

## 4. Discussion

In this study, we investigated the incidence, knowledge, attitude, and practices about HIV and HCV among migrant agricultural labourers residing in the Apulian ghettos. We found low levels of knowledge, notably regarding HCV, a favourable attitude toward these infections, and poor practices. The majority of migrants interviewed originated from Asia and Africa. Nearly every immigrant who joined was male and young (under 40 years of age), and many of them stated that they had been in the ghetto for roughly a year and a half out of their four years in Italy. A total of 117 migrants (38%) reported having regular employment contracts, whereas only 89 (29%) had a primary care physician. Ninety-five of the questioned migrants (30.7%) had previously been tested for HIV, whereas only three (0.9%) had been tested for HCV. Considering the modes of transmission, the level of knowledge of these diseases is low, particularly for HCV (approximately 30% of the examined population is aware of the mode of transmission, compared to 70% for HIV), and despite only four migrants testing positive, the vast majority did not use condoms during sexual encounters.

In our study, the incidence of HIV infection was 1.3% more than HIV incidence in Italy (0.6%) [12]. In the last decade, Apulia has reported approximately 161 HIV cases per year. In 2020, the incidence was 1.8 cases per 100,000 residents, while in 2019, it was 4.2 cases per 100,000 residents, likely because of the underdiagnosis and/or undernotification related to the COVID-19 pandemic. The highest incidence was found in the 30–39 age group [12], which is about the same age as the participants in our study, who were almost all under 40 years old.

In our country, the percentage of subjects infected with HCV reported by the main studies is about 2% of the general population [13]. In January 2020, it was estimated that there were approximately 30,000 people (an incidence of 0.76%) in the Apulian region with chronic active HCV infection who had not yet been treated with antiviral therapy [14]. The incidence among migrants in our study was higher (2.9%); this reflects evidence in the literature that highlights a higher HCV incidence among migrants than in that of the general population [15]. According to the European Centre for Disease Prevention and Control (ECDC), the risk of HIV infection and associated co-infections, such as HBV, HCV, and tuberculosis, is higher for migrants in the European Union and European Economic Area (EU/EEA). Indeed, in the West, migrants still represent half of all diagnosed individuals (over 47%); in Italy, the number of new diagnoses among migrants comprises about one-third of all new diagnoses in 2017 [16]. Migrant men who have sex with men, heterosexual migrant and ethnic minority men who engage in high-risk behaviors, and migrant women have all been identified as being among the groups most at risk for HIV. Higher HIV incidence among some migrant groups compared to the general population is linked to epidemiological patterns in countries of origin and also to higher exposure in countries of destination due to vulnerability and poor living conditions [17]. Several experiences have demonstrated that there are inequalities in access to healthcare services between migrants and the general population, as well as an increased risk of poor health status among migrants [18]. In addition, individuals may be excluded from the general health-care system due to a language barrier and conditions related to their poor living conditions as a result of their difficulty in finding a work or a regular contract, which impedes their inclusion in the social and health systems [11,19]. In our survey, only 89 migrants (29%) reported having a family doctor and thus primary access to care. According to the 2017 WHO Global Hepatitis Report, only 20% of HCV infected people are aware of their infection globally [20]. With regard to HIV, the Joint United Nations Programme on HIV/AIDS (UNAIDS)reports that 85% of all people living with HIV knew their status in 2021. In 2021, about 5.9 million people did not know that they were living with HIV [21]. Migrants are less likely to know if they are infected because the health care in their home countries isn’t good enough or because they live in remote areas in the country that they move to. However, not knowing about these diseases makes it harder to diagnose and treat them [11,22]. In addition, among the questioned migrants, nearly no one had been tested for HCV. Multiple studies have found a high frequency of this illness among migrants and stressed the susceptibility of travelers in their destination countries and throughout their migration path. Migrants who landed in western Sicily between 2015 and 2017 were provided early testing for the Hepatitis B virus (HBV), Hepatitis C virus (HCV), and human immunodeficiency virus (HIV). The acceptance percentage of 95.9% (2,751 migrants) reflected the favourable attitude revealed in our study. The rate of HCV was 0.9%, but the incidence of HIV was a staggering 2.2%. In particular, HIV infection was more prevalent among women who stayed in Libya for an extended period and were subjected to physical and/or sexual assault, highlighting one of the many reasons why migrants have a higher risk of contracting HIV and other illnesses [23]. Other studies reported a high prevalence of HIV among sub-Saharan African adults who interacted with the native population. Significant numbers were socially, legally, and economically disadvantaged, as seen by their unauthorised status, financial difficulties, and lack of secure housing [24,25], highlighting how living on the margins of society in poor living conditions increases the risk of disease and negative outcomes.

Therefore, it follows that travelling to Europe and living in a marginalised condition in Europe are two situations when there is a greater risk of contracting these infections.

In our study, the majority of migrants (96%) shared a stopover in Libya, and their journey from their country of origin to Italy lasted approximately one year on average. Most of the migrants who tested positive for HBV, HCV, or HIV in southern Italy between March 2019 and February 2020 and were sent to the Fondazione ARCA to register and apply for refugee status or a temporary residential permit were also ones who made the stopover in Libya. In this study, HCV and HIV infections were only found in migrants who had lived in Italy for more than 6–12 months. This could mean that the migrants probably got the infections after they moved to Italy and also that Libya, as suggested by several international organizations, is an indirect factor of violence [3]. Similarly, the migrants we screened have been in Italy for some time (about 4.8 years) and have been living in the ghetto for about 1.6 years on average; their travel from their country of origin to Italy lasted about 1 year on average. This emphasises the significance of ensuring access to prevention for migrant communities.

In line with our results, other surveys found low knowledge of therapy and a good attitude. In spite of having less knowledge regarding the effectiveness of therapy, migrants who took part in a French study were generally well-informed about HIV, but substantially less about hepatitis. Most of the participants did not know that the liver was the affected organ, or that the disease was transmittable by blood. This lack of information may be due to the fact that hepatitis is not considered a “plague” like HIV. Migrants’ acceptability of HIV and hepatitis testing was high [26], and other studies in migrants have suggested a high level of acceptability for screening for infectious diseases [27]. According to a German study, there are particular gaps in knowledge of HIV among younger, more recent migrants, those without regular access to the health care system, and those with a lower socio-economic status and a Muslim religion. Less than half of participants reported always using condoms with non-steady sexual partners [28]. The literature identifies a variety of social, economic, cultural, and legal issues that make immigrants and members of ethnic minorities more vulnerable to HIV infection. High-risk behaviours and limited access to healthcare are linked to increased susceptibility.

In addition, migrant men who have sex with men are an “invisible” group in HIV discussions. This invisibility is thought to contribute to their higher vulnerability to HIV infection. According to UK studies, men from sub-Saharan Africa report high levels of risky sexual behavior, yet they consider HIV prevention initiatives to be largely focused on women and children [29,30].

Dias et al. studied the role of mobility in the assumption of sexual risks and the acquisition of HIV among sub-Saharan African migrants living in two European cities. In this study, the overall percentage of those who had sex without condoms in the study sample was high (68.1%), confirming high levels of risky sexual behaviour among migrant populations. The specific risk factors associated with sex without condoms included being female, being over 30, traveling, having had the last sexual encounter with a regular partner, never having been tested for HIV, and having a non-reactive test result in the study. More than half of the travelers reported concurrency, i.e., having a regular partner in the host country while having other sexual partners abroad [31]. In addition to the high incidence of these infections among the migrant population and the gaps in linkage to care, there are several other reasons why this subgroup must be screened for viral hepatitis and HIV. Regarding HCV infection, it is known that chronic viral hepatitis has a protracted asymptomatic course during which infected persons are ignorant of their infection, up until the severe illness stage [32,33]. Each year, approximately 1.3 million deaths are caused by viral hepatitis, primarily due to chronic liver disease and its consequences [33]. Oral direct-acting antiviral (DAA) therapy for 8 to 12 weeks can easily eradicate HCV infection while lowering the risk of developing hepatocellular carcinoma and the progression to liver cirrhosis [34,35]. Considering that most of the mortality and medical costs were attributable to advanced liver disease, the early diagnosis and treatment of HCV are a very important issue for promoting public health.

Consequently, these categories must be addressed if viral hepatitis is to be eradicated in the European Economic Area or worldwide [11,36]. The World Health Organization has developed a set of goals for hepatitis elimination which include a 65% reduction in HCV-related deaths and a 90% reduction in HCV incidence by 2030 [37].

Furthermore, late HIV diagnosis is associated with an increased risk of morbidity and mortality, and may reduce the response to treatment; moreover, those who are diagnosed late are likely to utilise more healthcare resources; finally, late presentation increases the probability of transmission. Late presenters have a lower perceived risk of infection, are not routinely offered HIV testing, and are frequently from marginalised groups [38,39].

Twenty years of research demonstrate that HIV therapy is very efficient in preventing HIV transmission and that HIV-positive individuals with undetectable viral loads cannot transfer the virus sexually [40].

With the increased availability of antiretroviral medication (ART), worldwide and regional policies and guidelines emphasise the individual and public health advantages of HIV testing. In addition, they emphasise the need for early HIV detection and the link between testing and treatment, care, and support. However, not all nations, including those that recognise the heightened HIV risk of migratory communities, have clear HIV testing guidelines for these people [41]. There is evidence in the literature that HIV-HCV testing strategies in migrant populations are effective, including in terms of cost, identifying strategic advantages in rapid counselling and rapid testing [42,43], which could also be extended to education and prevention programmes for other sexually transmitted diseases.

We recognize that there are some limitations in our study: first of all, the small sample may not be representative of the entire ghetto population, but the difficulty of reaching this population nevertheless makes the data very interesting. Furthermore, the questionnaire used for the survey is not a validated questionnaire [44,45], but is in any case the result of a literature review.

## 5. Conclusions

Our findings show that agricultural migrant workers have a higher incidence of both HIV and HCV than the native population, as well as a lack of knowledge about both viruses, particularly HCV. We do not know whether the higher incidence of infection stems from the epidemiology of the countries of origin or whether infections contracted in Italy are also related to poor living conditions. The high willingness to participate in awareness-raising and screening programmes shown by migrants is a key element to being able to implement educational, screening, and prevention programs, and enable a greater diffusion of correct lifestyles. Of course, correct lifestyles cannot be detached from the environment in which people live. If this population lives in shanty towns/ghettos far from inhabited centers, isolated without water or sanitation (not by choice, but because they are exploited by the agromafia phenomenon), it is difficult to talk about lifestyles. It is unthinkable that people in Europe will be living in such exploitative conditions in 2023. It is urgent that a coordinated action between NGOs, politicians, universities, and volunteers is essential to highlight the issue of ghettos and their extremely poor living conditions and to take the most effective action possible to get those living in them out, as well as to ensure that such settlements no longer exist in a world that wants to truly call itself civilised.

## Figures and Tables

**Table 1 viruses-15-00249-t001:** Baseline sociodemographic characteristics of 309 agricultural migrant workers who participated in the KAP (knowledge, attitude and practices) survey and screening for HIV-HCV.

Variables	Frequency (n)	Percentage (%)
	Tot 309	
Gran Ghetto	69	22.3
Ghetto Pista	175	55.6
Casa Sankara	65	22.1
Male	272	88
Age (years) Mean (SD)	28.5	
18–30	166	53.7
31–45	124	40.1
>45	19	6.2
Marital status		
Single	201	65
Married	108	35
Educational status		
<8	98	31.7
>8	211	68.3
BMI		
>30	3	1
25–29.9	15	4.8
18.24.9	180	58.2
<18.5	101	36
Religion		
Muslim	256	82
Christian	54	18
African Continent	239	77.4
Asian Continent	70	22.6
**Screening results**		
HIV Positivity	4	1.3
HCV positivity	9	2.9
How long in Italy (mean, months)	58.3 (2–150)	-
How long in the ghetto (months)	19.3 (3–109)	-
Length of the travel (from country to Italy	12.2 (7–22)	-
Stopover in Lybia		
Yes	297	96

Do you have a regular contract for work		
yes	117	38

Do you have a family doctor		
Yes	89	29
Previous HIV test?		
yes	95	30.7
Previous HCV test?		
yes	3	0.9
Comorbidity?		
yes	49	15.8

**Table 2 viruses-15-00249-t002:** HIV-HCV Knowledge.

Questions	Strongly Disagree		Disagree		Neither Agree Nor Disagree		Agree		Strongly Agree	
	HIV	HCV	HIV	HCV	HIV	HCV	HIV	HCV	HIV	HCV
HIV/HCV can be transmitted through kissing a person living with HIV/HCV	77(25)	25(8)	108(35)	37(12)	77(25)	93(30)	31(10)	108(35)	15(5)	46(15)
HIV/HCV is transmitted through unprotected sex	46(15)	90(29)	68(22)	87(28)	31(10)	71(23)	124(40)	46(15)	71(23)	15(5)
With HIV treatment people can live a good quality of life and no longer be infectious/There is a HCV treatment that allows a definitive cure	46(15)	46(15)	77(25)	93(30)	31(10)	124(40)	93(30)	31(10)	62(20)	15(5)
HIV/HCV, if not treated, is transmitted from mother to child during birth	25(8)	62(20)	25(8)	109(35)	31(10)	46(15)	167(54)	71(23)	62(20)	22(7)
HIV/HCV is transmitted by contamination with infected blood, tattooing, syringe use	19(6)	108(35)	31(10)	68(22)	31(10)	71(23)	169(55)	49(16)	59(19)	43(14)
HCV causes liver cancer	65(21)		59 (19)		59(19)		62(20)		65 (21)	
With HIV treatment people can live a good quality of life and no longer be infectious.	31(10)		74(24)		53(17)		86(28)		56(18)	
Median Knowledge score	HIV 3 (IQR 1-4; maximum score: 7 points) HCV 1 (IQR 0-3; maximum score: 6 points)									

**Table 3 viruses-15-00249-t003:** Attitude about HIV and HCV.

Questions	Strongly Disagree		Disagree,		Neither Agree Nor Disagree		Agree		Strongly Agree	
	HIV	HCV	HIV	HCV	HIV	HCV	HIV	HCV	HIV	HCV
I would take HIV/HCV treatment	0 (0)	0(0)	0(0)	3(1)	25(8)	15(5)	170(55)	185(60)	114(37)	105(34)
Will not maintain friendship if a friend with HIV/HCV infection	154(50)	127(41)	124(40)	136(44)	12(4)	31(10)	15(5)	15(5)	3(1)	0(0)
Will not host an individual living with HIV/HCV at home	139(45)	124(40)	71(23)	124(40)	77(25)	46(15)	31(10)	9(3)	21(7)	2(6)
I would be in favour of education compains on HIV/HCV	0(0)	0(0)	0(0)	0(0)	3(1)	0(0)	263(85)	247(80)	46(15)	62(20)
I would be in favour of HIV/HCV screening compains	0(0)	0(0)	0(0)	0(0)	0(0)	0(0)	238(77)	263(85)	71(23)	46(15)
Attitude median score	5 (IQR 5-6; maximum score:5 points) on both HIV-HCV education compains									

**Table 4 viruses-15-00249-t004:** HIV and HCV Practices.

Questions	Strongly Disagree	Disagree	Neither Agree nor Disagree	Agree	Strongly Agree
I always have sex with a condom	179(58)	37(12)	77(25)	6(2)	9(3)
I never used siringes already used	3(1)	3(1)	6(2)	226(73)	71(23)
I recommend the use of condoms to my friends during sex.	62(20)	93(30)	93(30)	15(5)	15(5)
Sex under influence of drugs or alcohol	185(60)	99(32)	15(5)	6(2)	3(1)
In the last 3 months did you have a dangerous sexual intercourse ?	77(25)	15(5)	124(40)	62(20)	31(10)

**Table 5 viruses-15-00249-t005:** Factors associated with a low knowledge of HCV.

Characteristics	Univariate Analysis OR	Multivariate Analysis Adj-O.R.
Age (1 ys)	1.02 (0.98–1.04)	-
Male	1.51 (1.42–2.02)	1.72 (1.28–1.92)
Educational status < 8	1.80 (1.50–2.00)	2.09 (1.28–2.21)
Single	1.14 (1.08–1.78)	1.63 (1.20–2.03)
BMI > 25	0.34 (0.10–1.10)	1.24 (0.69–1.45)
>12 months in the ghetto	1.85 (1.35–2.45)	1.95 (1.25–2.55)
African continent	1.21 (1.13–1.70)	1.43 (1.28–2.01)
have regoular contract for work	0.51 (0.41–1.10)	0.64 (0.39–0.83)
family doctor	0.25 (0.10–0.40)	0.24 (0.18–0.73)
Previous HIV/HCV test performed	0.34 (0.10–1.10)	0.59 (0.10–0.90)

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
