# Peer review of "HIV-HCV Incidence in Low-Wage Agricultural Migrant Workers Living in Ghettos in Apulia Region, Italy: A Multicenter Cross Sectional Study"

_viruses, 2023, doi:10.3390/v15010249_

Round 1

Reviewer 1 Report (Previous Reviewer 3)

The authors did not review my two main questions that I posed earlier, so I repeat my questions:

1. The sample is purely for convenience, and a sample power test was not performed to ensure its originality and relevance. Thus, it is impossible to state that these results represent this population;

Please perform a test to assess the power of this sample to represent the population. Basic software like G-power does it in less than 5 minutes.

2. The Knowledge, attitudes and practices scale does not present any validation process that attests its applicability. The validity and/or calibration of an instrument is conditioned not only to content validation (as the authors did), but to the number of times it is tried and the analytical procedures, including multivariate statistics. There are several studies that reinforce the need for a combination with more consistent and robust analyses, such as criteria and constructs, in order to gather more evidence about validity. Reliability is also something that should be added to the results: See: Souza AC, Alexandre NMC, Guirardello EB. Psychometric properties in instruments evaluation of reliability and validity. Epidemiol Serv Saude. 2017 Jul-Sep;26(3):649-659. English, Portuguese. doi: 10.5123/S1679-49742017000300022. PMID: 28977189; https://doi.org/10.1590/0102-311X00143613; Reichenheim ME, Hökerberg YH, Moraes CL. Assessing construct structural validity of epidemiological measurement tools: a seven-step roadmap. Cad Saude Publica. 2014 May;30(5):927-39. doi: 10.1590/0102-311x00143613. PMID: 24936810. Content validation is an important step, but it should be added to other procedures and even in a more representative sample for the convergence of objectives, that is, that the instrument measures what it really intends and or was intended to measure.

Please provide some test of feasibility of the questionnaire, either as judges or with the target population, for example the CVI (content validation index).

Author Response

 Response: Many thanks for your comments. Thanks to your comments, we believe the manuscript has greatly improved and thank you for approving the publication of the paper. We modified the paper following your suggestions on several points and a native English speaker revised the paper. About simple size: Precise data regarding the incidence of HIV and HCV among low-wage agricultural migrant workers living in ghettos are not available. However, hypothesizing a prevalence of 2% , based on other studies made in similar settings, with an alpha-error of 5% and a power of 80%, the recommended sample size was 31 participants.

Response: Thank you, we add this in limitation section

“We recognize some limitations in our study: first of all, the small sample may not be significant of the entire ghetto population, but the difficulty of reaching this population nevertheless makes the data very interesting. Furthermore, the questionnaire used for the survey is not a validated questionnaire [44-45],  but is in any case the result of a literature review.”

Ref 44. Souza AC, Alexandre NMC, Guirardello EB. Psychometric properties in instruments evaluation of reliability and validity. Epidemiol Serv Saude. 2017 Jul-Sep;26(3):649-659. English, Portuguese. doi: 10.5123/S1679-49742017000300022. 

Ref 45. Reichenheim ME, Hökerberg YH, Moraes CL. Assessing construct structural validity of epidemiological measurement tools: a seven-step roadmap. Cad Saude Publica. 2014 May;30(5):927-39. doi: 10.1590/0102-311x00143613

In addiction, also in our previous published survey,  see

- Di Gennaro F, Occa E, Chitnis K, Guelfi G, Canini A, Chuau I, Cadorin S, Bavaro DF, Ramirez L, Marotta C, Cotugno S, Segala FV, Ghelardi A, Saracino A, Periquito IM, Putoto G, Mussa A. Knowledge, attitudes and practices on cholera and water, sanitation, and hygiene among internally displaced persons in Cabo Delgado Province, Mozambique. Am J Trop Med Hyg. 2022 Dec 12:tpmd220396. doi: 10.4269/ajtmh.22-0396.

- Di Gennaro F, Murri R, Segala FV, Cerruti L, Abdulle A, Saracino A, Bavaro DF, Fantoni M. Attitudes towards Anti-SARS-CoV2 Vaccination among Healthcare Workers: Results from a National Survey in Italy. Viruses. 2021 Feb 26;13(3):371. doi: 10.3390/v13030371. 

- Di Gennaro F, Marotta C, Amicone M, Bavaro DF, Bernaudo F, Frisicale EM, Kurotschka PK, Mazzari A, Veronese N, Murri R, Fantoni M. Italian young doctors' knowledge, attitudes and practices on antibiotic use and resistance: A national cross-sectional survey. J Glob Antimicrob Resist. 2020 Dec;23:167-173.

we don’t use the content validation because if the context or reference population is unique, it is difficult to apply it, so as a decision of all authors we did not.

I repeat we put this point, correctly with your observation, in the limitations  section of the paper.

Reviewer 2 Report (Previous Reviewer 2)

The revised manuscript has been improved as request. 

Author Response

Many thanks for your comment. Thanks to your comments, we believe the manuscript has greatly improved and thank you for approving the publication of the paper. 

Round 2

Reviewer 1 Report (Previous Reviewer 3)

Dear authors, thank you for your sincerity in exposing the limitations.

This manuscript is a resubmission of an earlier submission. The following is a list of the peer review reports and author responses from that submission.

Round 1

Reviewer 1 Report

the title should be modified by adding the word Italy at the end

Reviewer 2 Report

Dear Authors

Thank you for the opportunity to revise your interesting paper titled “Low HCV knowledge in HIV-HCV screening among low-wage agricultural migrant workers living in ghettos in Apulia.” – “viruses-2115968

The manuscript treats an interesting field but some criticisms need to be addressed before to consider it for publication. Below I mentioned major criticisms:

Title

-       The title should be more exhaustive and in line with the primary aim of the study. The principal aim is “prevalence” or “knowledge”?

-       Pag 1 Line 27-29 you mentioned the “…explore prevalence of HIV and HCV ………and their knowledge….”

-       Pag. 2 Line 84-86 you wrote the “The primary objectives of the study are to ascertain the prevalence of HIV and HCV among migrant agricultural labourers, as well as their knowledge”

-       Please check it in the title also, in the title, should be indicate the method (i.e.: cross sectional study?)

Method section

-       The method section should be more exhaustive, the period of data recruitment should be mentioned, you reported “Between January 10, 2020, and April 20, 2021,” in the “results section”. Please check it.

-       Should be clarified how medical resident in infectious diseases and a nurse administered the questionnaires through a face-to-face interview. Please check it.

-       Please insert “ethical considerations” sub-section.

Conclusion

-       Conclusions section – this section should be start with the principal aim of the study (prevalence not knowledge).

Reviewer 3 Report

Thank you very much for the opportunity to review your manuscript. I think the idea is original and relevant, but there are methodological errors that prevent its progress

1. The sample is purely for convenience, and a sample power test was not performed to ensure its originality and relevance. Thus, it is impossible to state that these results represent this population;

2. The Knowledge, attitudes and practices scale does not present any validation process that attests its applicability. The validity and/or calibration of an instrument is conditioned not only to content validation (as the authors did), but to the number of times it is tried and the analytical procedures, including multivariate statistics. There are several studies that reinforce the need for a combination with more consistent and robust analyses, such as criteria and constructs, in order to gather more evidence about validity. Reliability is also something that should be added to the results: See: Souza AC, Alexandre NMC, Guirardello EB. Psychometric properties in instruments evaluation of reliability and validity. Epidemiol Serv Saude. 2017 Jul-Sep;26(3):649-659. English, Portuguese. doi: 10.5123/S1679-49742017000300022. PMID: 28977189; https://doi.org/10.1590/0102-311X00143613; Reichenheim ME, Hökerberg YH, Moraes CL. Assessing construct structural validity of epidemiological measurement tools: a seven-step roadmap. Cad Saude Publica. 2014 May;30(5):927-39. doi: 10.1590/0102-311x00143613. PMID: 24936810. Content validation is an important step, but it should be added to other procedures and even in a more representative sample for the convergence of objectives, that is, that the instrument measures what it really intends and or was intended to measure.

3. Analysis models are confusing. It is not clear which parameters for each variable to enter the adjusted models, much less how the adjustments were made.